# Characteristics of the Complete Plastid Genome Sequences of the Monotypic Genus *Dodecadenia* (Family: Lauraceae) and Its Phylogenomic Implications

**Chao Liu** [1,2] , **Huanhuan Chen** [1,2], **Jian Cai** [1,2], **Xiangyu Tian** [3], **Lihong Han** [1,2],*** and **Yu Song** [4,5],*****

1. College of Biological Resource and Food Engineering, Qujing Normal University, Qujing 655011, China
2. Yunnan Engineering Research Center of Fruit Wine, Qujing Normal University, Qujing 655011, China
3. School of Life Sciences, Zhengzhou University, Zhengzhou 450001, China
4. Key Laboratory of Ecology of Rare and Endangered Species and Environmental Protection (Ministry of Education), Guangxi Normal University, Guilin 541004, China
5. Guangxi Key Laboratory of Landscape Resources Conservation and Sustainable Utilization in Lijiang River Basin, Guangxi Normal University, Guilin 541004, China
* Correspondence: hanlihong9527@126.com (L.H.); songyu@xtbg.ac.cn (Y.S.)

**Abstract:** As one of a dozen monotypic genera in the family Lauraceae, the systematic position of *Dodecadenia* Nees remains controversial. Here, two complete plastomes of *Dodecadenia grandiflora* Nees were sequenced. The two plastid genomes, with the length of 152,659 bp and 152,773 bp, had similar quadripartite structure. Both consisted of one large single-copy (LSC) region with 93,740 bp and 93,791 bp, one small single-copy region (SSC) with 18,805 bp and 18,846 bp, and a pair of inverted repeats (IR) regions with 20,057 bp and 20,068 bp. A total of 128 genes were annotated for the *D. grandiflora* plastid genomes (plastomes), which included 84 protein-coding genes (PCGs), 36 tRNA genes and eight rRNA genes. Codon usage analysis of the *D. grandiflora* plastomes showed a bias toward A/U at the third codon. A total of 122 RNA editing events were predicted, and all codon conversions were cytosine to thymine. There were 30/36 oligonucleotide repeats and 89/94 simple sequence repeats in these two plastomes of *D. grandiflora*. Based on 71 plastomes, both Bayesian and maximum likelihood phylogenetic analyses showed that *D. grandiflora* are nested among the species of *Litsea* Lam. together with *Litsea auriculata* Chien et Cheng and suggested that the monotypic genus *Dodecadenia* Nees should be revised. In addition, the highly variable loci *trnG* intron and *ycf3-trnS* could be used as excellent candidate markers for population genetic and phylogenetic analyses of *D. grandiflora*.

**Keywords:** *Dodecadenia*; Lauraceae; plastid; phylogenetic; repeats; RNA editing site





## 1. Introduction

The monotypic genus in the family Lauraceae includes *Dahlgrenodendron* J.J.M.van der Merwe and A.E.van Wyk, *Dodecadenia* Nees, *Eusideroxylon* Teijsm. and Binn., *Hexapora* Hook.f. *Hypodaphnis* Stapf, *Iteadaphne* Blume, *Paraia* Rohwer, H.G.Richt. and van der Werff, *Parasassafras* D.G.Long, *Potoxylon* Kosterm., *Sinopora* J.Li, N.H.Xia & H.W.Li, *Sinosassafras* H.W.Li, and *Umbellularia* Nutt. (http://foc.iplant.cn/, accessed on 20 June 2022). *Dodecadenia* comprises *D. grandiflora* Nees 1831 and belongs to the tribe Laureae, mainly distributed in Himalayan and Hengduan Mountains (Bhutan, India, Myanmar, Nepal, and Yunnan, Tibet and Sichuan of China) [1]. The tribe Laureae, also known as the *Litsea* complex, is composed approximately of 500 species and 10 genera: *Actinodaphne* Nees, *Adenodaphne*, *Dodecadenia*, *Iteadaphne*, *Laurus* L., *Lindera* Thunb, *Litsea* Lam., *Neolitsea* Merr., *Parasassafras* and *Sinosassafras* H.W.Li [2–7]. In the previous classification system of Lauraceae, the *Litsea* complex is characterized by consistent morphological features of racemose inflorescences [1,8]. In Kostermans' system, the tribe Litseeae was divided

into 4-celled anther subtribe Litseineae and 2-celled anther subtribe Lauriineae, including *Litsea / Neolitsea* and *Lindera / Laurus*, respectively. *Dodecadenia* and *Iteadaphne* were treated as subgenera in the *Litsea* and *Lindera*, respectively [9]. In the process of the transformation of the racemose inflorescence into a pseudo-umbel, a single flower occurs in the involucre of *Dodecadenia* and *Iteadaphne* [2]. The *Litsea* complex is basically characterised by introrse pollen sacs, and latrorse pollen sacs occurring in the *Iteadaphne* and *Dodecadenia* [2]. However, androecial characteristics are often variable, even within genera, and should not be used in the classification of Lauraceae [8]. The number of anther cells is not a valuable characteristic for phylogenetic identification of core Lauraceae [10]. Generic delimitation within the *Litsea* complex has traditionally been problematic [11]. Furthermore, a major disparity exists between molecular data and morphology-based classifications, probably due to the complex interpretation of the morphological characteristics of Lauraceae [12,13]. In order to distinguish the *D. grandiflora* and *Litsea* species and reconstruct their systematic relationships, a molecular approach is of considerable interest.

The earlier reported molecular markers for the Lauraceae were the *trnL-trnF*, *trnT-trnL*, *psbA-trnH*, and *rpll6* regions of plastid, and the 26S and 5.8S regions of ribosomal DNA (rDNA) [3]. Based on the *matK* of the plastid genome (plastome) and the internal transcribed spacer (ITS) of rDNA of the nuclear genome, Li et al. [11] revealed that most genera of the *Litsea* complex are polyphyletic and display conflicts between *matK* and ITS. In the *Litsea* complex, *D. grandiflora* has previously been retrieved as the sister to *Laurus nobilis* L., with 89% support in their *matK* data, and to *Litsea glutinosa* (Lour.) C. B. Rob. and *Litsea* sect. Cylicodaphne (BS = 78%) in the *matK* and ITS combined data [11]. Another study [12], using ITS and the external transcribed spacer (ETS) sequences with 71 species, constructed a phylogenetic tree in order to settle the positions of core Laureae, which indicated the close relationships between *D. grandiflora* and *Litsea elongata* (Wall. ex Nees) Benth. et Hook. f., *Litsea yaoshanensis* Yang et P. H. Huang, and *Litsea acutivena* Hay. Fijridiyanto and Murakami [4] performed the phylogenetic relationships of the tribe Laureae based on *matK* and ITS regions and found that several species of *Litsea* and *Lindera* were nested with each other; *Actinodaphne* and *Neolitsea* were monophyletic with a close relationship.

Due to its high conservation, few recombination incidents, low nucleotide replacement rates, and rapid evolution rate, the plastome has become an ideal tool for studying the phylogenetic relationships among species [14,15]. As more and more plastomes are decoded, phylogenetic analyses based on plastome have become increasingly popular, which provides an effective solution for solving systematic problems [16–18]. Compared to DNA barcodes or specific regions, plastome data provided a modest increase in discrimination in Lauraceae [13]. Based on plastome sequences, Lauraceae was divided into nine clades and *Laurus–Neolitsea* clade formed the *Litsea* complex [6]. Indeed, a large number of plastomes of Lauraceae have been released [6,15,19,20], while the plastid genome of *D. grandiflora* is not available yet.

*Dodecadenia grandiflora* is a terrestrial evergreen species used as an antihyperglycemic agent by the traditional medical practitioners [21]. Four compounds, two phenylpropanoyl esters of catechol glycosides and two lignane bis esters, extracted from the leaves of *D. grandiflora* showed significant antihyperglycemic activity in streptozotocin-induced diabetic rats [21]. In the present study, the complete plastomes of two *D. grandiflora* samples were de novo assembled using Illumina sequencing technology, and the features of the plastomes were fully elucidated. Combined with the other 69 plastomes of the Lauraceae species from a public database, the genealogical relationships between *D. grandiflora* and other species were elucidated. Our aims were to investigate the structural pattern of complete plastome, and to determine phylogenetic status for *D. grandiflora* species based on complete plastome sequences. Our results would enrich the plastome database of Lauraceae and provide resources for utilizing *D. grandiflora* in future studies.

## 2. Materials and Methods

### 2.1. Plant Materials and Plastome Sequencing

Fresh leaves from two wild *D. grandiflora* seedlings were collected and identified by Dr. Yu Song (Guangxi Normal University): *D. grandiflora* I from Yongsheng County Yunnan and *D. grandiflora* II from Jilong County Tibet, China. The specimens were deposited at the herbarium of Guangxi Normal University with the archival number of BRG-SY37528 and BRG-SY61123. Total genomic DNA was extracted from the leaf using the CTAB method [22]. The 150 bp paired-end reads were produced to construct libraries by next-generation sequencing platform on an Illumina NovaSeq 6000 at Novogene Bioinformatics Technology Co., Ltd. (Tianjin, China). Approximately 3.4 Gb of raw data were yielded. The qualities of the clean reads were checked with FastQC v0.11.8.

### 2.2. Plastome Assembly and Annotation

The sequencing adapters and low-quality reads were filtered, and the circular plastomes were assembled de novo using GetOrganelle v1.7.0 [23]. Average read coverage of 95× of plastomes were assembled. The plastomes were annotated with GeSeq (https://chlorobox.mpimp-golm.mpg.de/geseq.html, accessed on 4 February 2022) [24] and Geneious v8.0.2 [25]. The map of annotated *D. grandiflora* plastome was generated with CHLOROPLOT software (https://irscope.shinyapps.io/Chloroplot/, accessed on 4 February 2022) [26]. The well-annotated plastomes of *D. grandiflora* was submitted to the public GenBank database under the accessions ON931229 (*D. grandiflora* I) and ON931230 (*D. grandiflora* II).

### 2.3. Sequence Analysis

The long repeats of the *D. grandiflora* plastomes, including the forward, reverse, palindrome, and complement types, were detected by the online REPuter program (https://bibiserv.cebitec.uni-bielefeld.de/reputer, accessed on 4 February 2022) [27] with a minimum repeat size of 30 bp and a hamming distance of 3. In addition, simple sequence repeats (SSR) were detected via MISA (https://webblast.ipk-gatersleben.de/misa/, accessed on 4 February 2022) [28] by setting the minimum number of repeats to 10, 5, 4, 3, 3 and 3 for mononucleotide, di-, tri-, tetra-, penta- and hexa-, respectively. The relative synonymous codon usage (RSCU) and amino acid frequencies were calculated using EMBOSS (http://emboss.toulouse.inra.fr/, accessed on 4 February 2022) and MEGA X [29] with default parameters. The RNA editing sites of 35 protein-coding genes (PCGs) in *D. grandiflora* were predicted by PREP-Cp program (http://prep.unl.edu/, accessed on 4 February 2022) [30].

### 2.4. Phylogenetic Analysis

To determine the phylogenetic status of *D. grandiflora* in the Lauraceae family, a total of 71 Lauraceae species plastomes were performed to construct phylogenetic trees (Table S1). We downloaded plastome sequences of 66 taxa across the *litsea* complex which correspond to seven genera (*Actinodaphne*, *Iteadaphne*, *Laurus*, *Lindera*, *Litsea*, *Neolitsea*, and *Parasassafras*) and three outgroup species (*Cinnamomum camphora*, *Cinnamomum micranthum*, and *Sassafras albidum*). These sequences were aligned by MAFFT v7 (https://mafft.cbrc.jp/alignment/server/, accessed on 4 February 2022) [31] and manually edited by BioEdit v7.0.9. Bayesian inference (BI) analyses were conducted with MrBayes v3.2 [32] for one million generations and tree sampling every 200 generations. Maximum-Likelihood (ML) analysis were performed using IQ-TREE v2.1.1 [33] with the GTR + F + R2 model and a bootstrap value (BS) of 1000.

### 2.5. Sequence Divergence and Comparative Genome Analysis

The plastomes pairwise alignments and sequence divergence between the *D. grandiflora* and those of 13 closely related species were analyzed using the mVISTA program (https://genome.lbl.gov/vista/mvista/submit.shtml, accessed on 4 February 2022) [34]

in Shuffle-LAGAN mode with *Litsea auriculata* as the reference. Gene order comparison of newly assembled *D. grandiflora* plastomes were performed using the Mauve plugin in Geneious v8.0.2 [25]. The nucleotide diversity (Pi) of the plastome sequences was calculated using DnaSP v6.10 [35] with a window length of 600 bp for sliding window with a step size of 200 bp.

## 3. Results

### 3.1. General Feature of the Plastome

The newly assembled plastomes of *D. grandiflora* had a quadripartite structure forming a circular molecule, and the size of the two genomes were 152,659 bp and 152,773 bp in length (Figure 1). The LSCs were 93,740 bp and 93,791 bp, the SSCs were 18,805 bp and 18,846 bp, and a pair of inverted repeats were 20,057 bp and 20,068 bp. The two genomes of *D. grandiflora* showed variation of 51, 41 and 11 bp in LSC, SSC and IR to some extent. The overall guanine-cytosine (GC) contents were 39.2% and 39.1%, respectively. The GC contents were unevenly distributed in different regions of the plastome, with 37.9%, 34.0%, and 44.4% for the LSC, SSC, and IR regions, respectively. In total, 128 genes were annotated in the plastome of *D. grandiflora*, including 84 PCGs, 36 tRNA genes and eight rRNA genes. Among 113 unique genes, 15 genes were duplicated in the IR regions, including four rRNA genes, six tRNA genes and five protein coding genes. A total of 14 PCGs and eight tRNA genes, and two genes (*ycf3* and *clpP*) possess two introns.

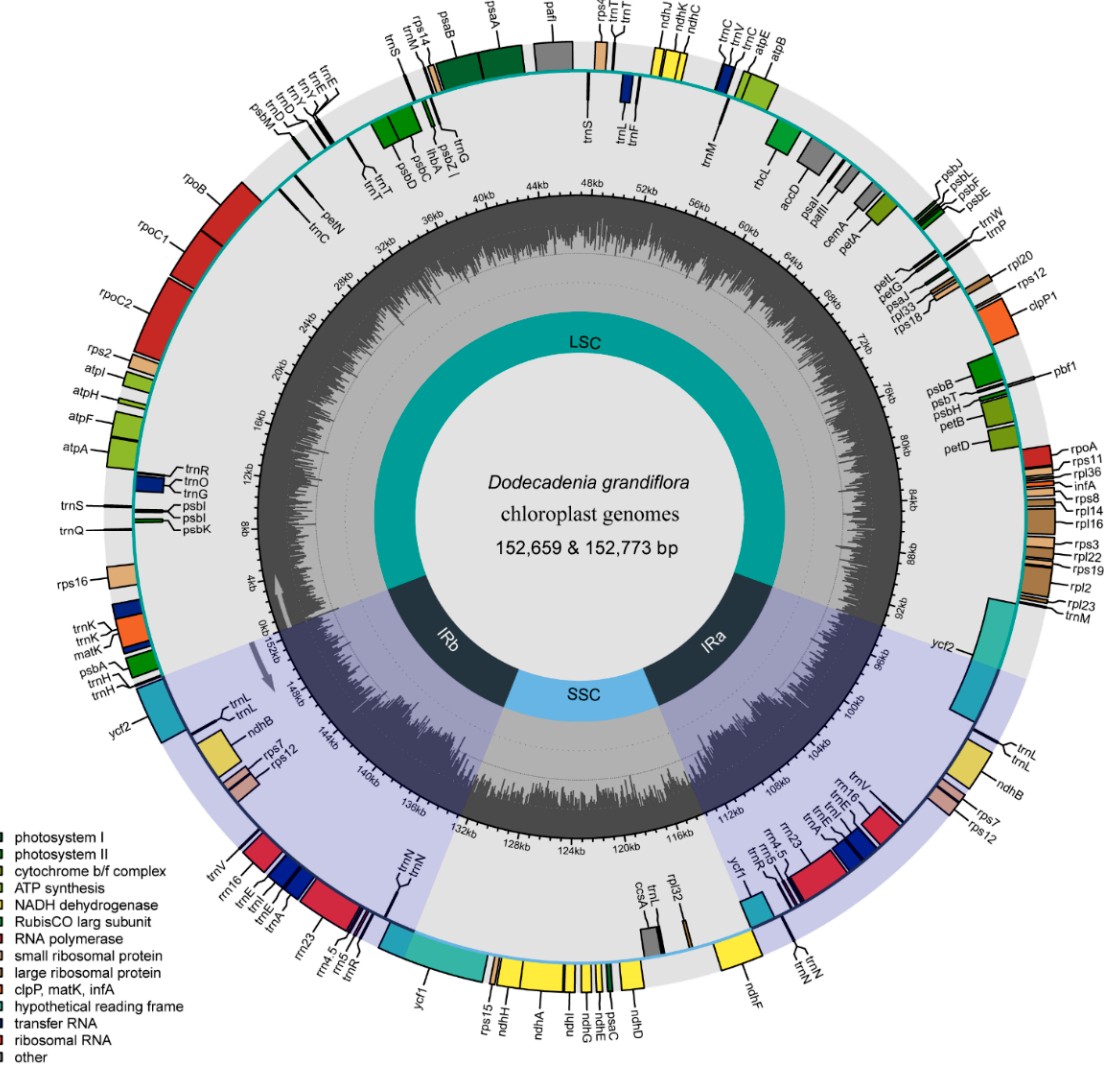

**Figure 1.** Gene map of *D. grandiflora* plastomes.

### 3.2. Amino Acid Frequency and Codon Usage Analysis

The two genomes of *D. grandiflora* showed similar patterns of amino acid frequency and codon usage. The results showed that the most abundant amino acids encoded are leucine (Leu, 10.27%), isoleucine (Ile, 8.53%), serine (Ser, 7.83%), and glycine (Gly, 7.11%). Least represented in the plastomes examined were cysteine (Cys, 1.18%) and tryptophan (Trp, 1.72%) (Figure S1). The RSCU of 31 codons was greater than 1.0. The count of preferred codons ending with A, U, C, G were 13, 16, one, and one, which showed a bias toward A and U at the third codon usage. Furthermore, the highest RSCU value was AGA (1.80) in arginine (Arg), and the lowest was AGC (0.34) in Ser (Figure 2, Table S2).

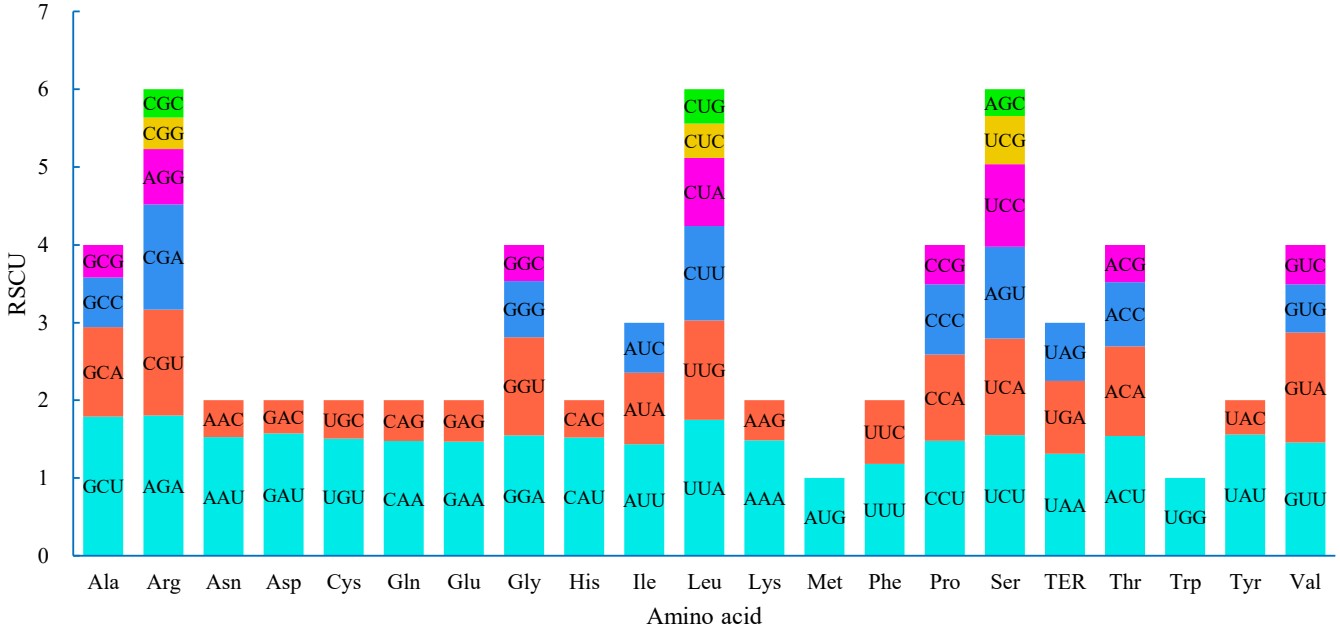

**Figure 2.** Codon usage patterns of *D. grandiflora* plastome. The y-axis represents the relative synonymous codon usage whereas x-axis represents the codons.

### 3.3. RNA Editing Sites Analysis

The RNA editing analyses showed consistent results with respect to genes and the position of editing sites in the PCGs of the two *D. grandiflora* genomes. A total of 122 putative RNA editing sites were identified using the PREP-CP software, which were distributed in 31 PCGs (Table S3). All the RNA editing nucleotide changes were from cytidine (C) to thymine (T), and 84 codons were substituted in second nucleotide position (68.9%) and 38 codons were substituted in first nucleotide position (31.1%). All amino acids showed only one type of nucleotide conversion position except Proline, which changed at the first nucleotide position converting proline (Pro) to Ser and phenylalanine (Phe) and changed at second nucleotide converting Pro to Leu. The highest number of RNA editing sites was observed in *ndhB* (15 sites), followed by *ndhD* (13 sites), and *rpoB* (12 sites) genes. The post-transcriptional substitutions of Arg to Cys, Arg to tryptophan, histidine to tyrosine, Leu to Phe, Pro to Phe, and Pro to Ser amino acids changes occurred due to the first codon positions, while the alanine to valine, Pro to Leu, Ser to Leu, Ser to Phe, threonine (Thr) to Ile, and Thr to methionine changes occurred due to the second codon positions (Figure 3). Of the 122 RNA editing sites, 70 changed the encoded amino acid from polar to apolar, and the most abundant amino acid change (42) was the Ser to Leu (Figure 3).

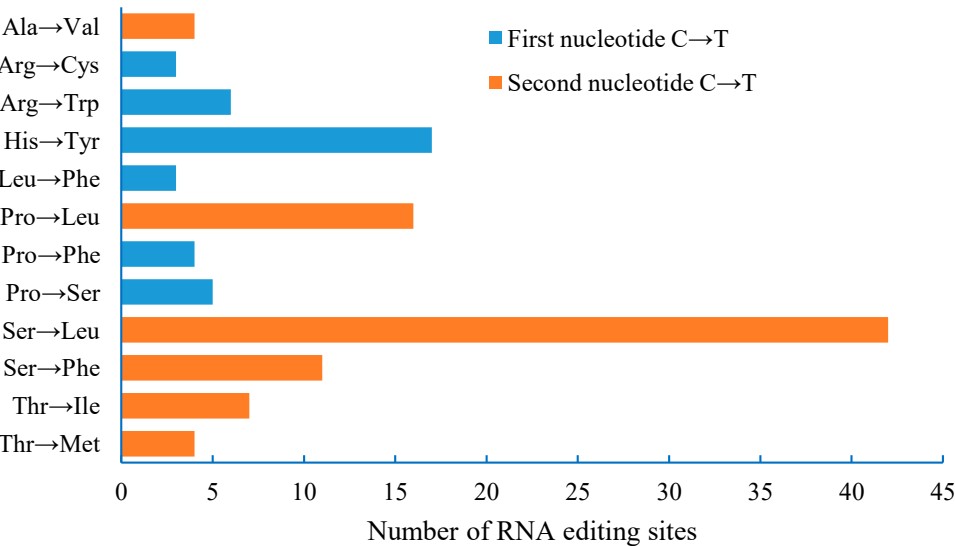

**Figure 3.** The prediction of RNA editing sites number and amino acid changes.

### 3.4. Repeats Analysis

Thirty/thirty-six long repeats (>30 bp) were predicted in two genomes of *D. grandiflora*, and the number of palindromic (P) was 11/14, the number of forward (F) was 11/14, the number of both reverse (R) was six, and the number of both complement (C) was two. Most of the repeat ranged from 30 to 39 bp in length and repeat number of 41 bp and 48 bp was one. A total of 8/10 long repeats were identified in LSC region, 13/17 were in IR, two were in SSC, and seven were distributed in the junctions, among which 1/0 were present in the LSC/SSC region, and 6/7 in the LSC/IR region (Figure 4, Table S4).

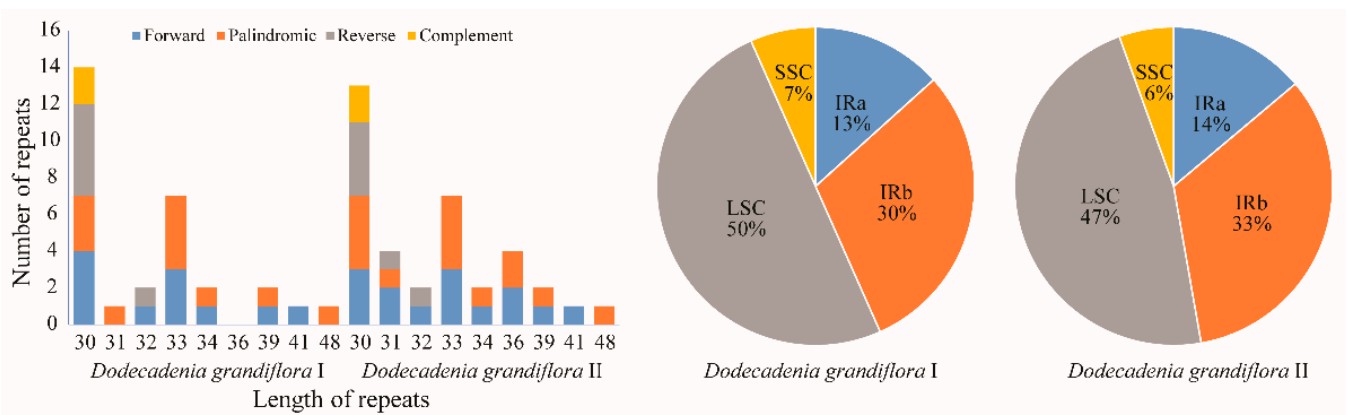

**Figure 4.** Number and distribution of long repeats in two plastomes of *D. grandiflora*.

The number of SSRs of plastome of *D. grandiflora* was detected using MISA (Figure 5). A total of 89/94 SSRs were detected in two genomes of *D. grandiflora*. Among these SSRs, the mononucleotide repeat units were the most identified SSRs, and 66/71 for mono nucleotide repeats were identified, all of them belong to A or T base repeats. while AT/TA, AAT/TAA, and AAAT/ATTT repeats were found most in the di-, tri-, and tetranucleotide types, respectively. There were 68/73, 17, and four SSR repeats distributed in LSC, SSC, and IRs of both genomes (Table S5). These SSRs may be potential specific molecular markers in genetic diversity and phylogenetic studies for *Dodecadenia*.

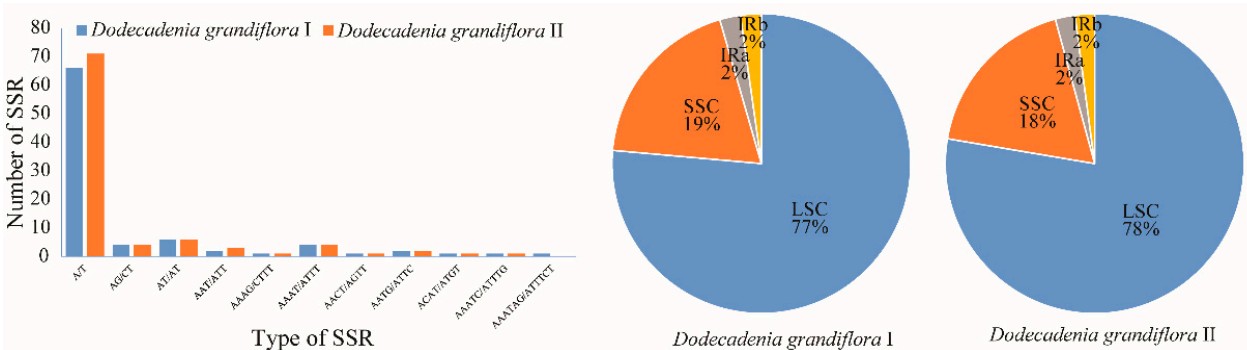

**Figure 5.** Number and distribution of SSRs in two plastomes of *D. grandiflora*.

### 3.5. Phylogenetic Analyses

Seventy-one plastomes of Lauraceae were used to examine the phylogenetic status of *D. grandiflora*. The topology of the phylogenetic trees produced with BI and ML methods were similar and topologically congruent and highly supported based on our complete plastome sequences. The relationship of the tribe Laureae clustered in a monophyletic group. The phylogenetic tree revealed that all species of the *Litsea* complex formed four strongly supported major clades (Figure 6). Five *Lindera* species (*Lindera glauca* (Siebold & Zucc.) Blume, *Lindera angustifolia* Cheng, *Lindera fragrans* Oliv., *Lindera nacusua* (D. Don) Merr., and *Lindera communis* Hemsl.) formed clade I. In clade II, two *Laurus* (*Laurus azorica* (Seub.) Franco and *Laurus nobilis*), two *Litsea* (*Litsea acutivena*, and *Litsea glutinosa*), and *Lindera megaphylla* Hemsl. formed a monophyletic group that was sister to the other six *Lindera* and nine *Litsea* species. Clade III comprised two monophyletic groups, one group was two *Lindera* and four *Litsea* species, another group contained *D. grandiflora* and the other 13 *Litsea* species. *Dodecadenia grandiflora* and *Litsea auriculata* grouped together in the tree with a strong support (BI = 1, BS = 100). In clade IV, eight *Lindera* species, eight *Neolitsea* species, four *Actinodaphne* species, *Iteadaphne caudata* (Nees) H. W. Li, and *Parasassafras confertiflorum* (Meisner) D. G. Long formed a monophyletic group that was sister to clade III.

### 3.6. Comparative Plastome Sequence Divergence and Hotspots Regions

A global sequence alignment of two *D. grandiflora* and 13 *Litsea* species plastomes in Clade IIIb were compared to determine interspecific divergence by mVISTA online software with *Litsea auriculata* as the reference. The results showed that these closely related species had few differences in gene sequence and content (Figure 7). Collinearity detection showed no gene rearrangements within the 15 plastomes (Figure S2). The divergences were more stable in the coding regions than the non-coding regions, and the sequences were more conserved in IR than LSC and SSC.

The nucleotide variability (Pi) values in Lauraceae plastome were calculated using DnaSP. The analysis indicated that the LSC and SSC exhibited higher Pi in comparison to IRs (Figure S3), and the most diverse regions were the intergenic spacers. The Pi values across the 15 plastomes varied from 0 to 0.0149, with an average value of 0.0031. Three highly variable loci *trnG*-intron, *trnC-petN*, and *ycf3-trnS* (Pi $\geq$ 0.0095) were located in the LSC, and five loci *ndhF*, *ndhF-rpl32*, *rpl32-trnL*, *rps15-ycf1*, and *ycf1* (Pi $\geq$ 0.0095) were located in the SSC (Figure S3a). The Pi values among the 68 *Litsea* complex plastomes vary from 0 to 0.0159, with a mean value of 0.0038 (Figure S3b). Six variable loci (Pi > 0.0095) were identified in the LSC region (*psbK-psbI*, *psbI-trnS*, *trnS-trnG*, *trnC-petN*, *psbM-trnD*, and *ycf2*) and five loci in the SSC region (*ndhF*, *ndhF-rpl32*, *rpl32-trnL*, *rps15-ycf1*, and *ycf1*). Additionally, six highly variable regions (*petA-psbJ*, *ycf1-ndhF*, *ccsA-ndhD*, *ndhH-rps15*, *rps15-ycf1*, and *ycf1*, Pi > 0.0095) and 325 mutation sites were identified between the two *D. grandiflora* plastomes, including ten microinversions, 77 indels, and 238 substitutions. Because of indels, *ndhF* of *D. grandiflora* II had an 81 bp longer coding sequence.

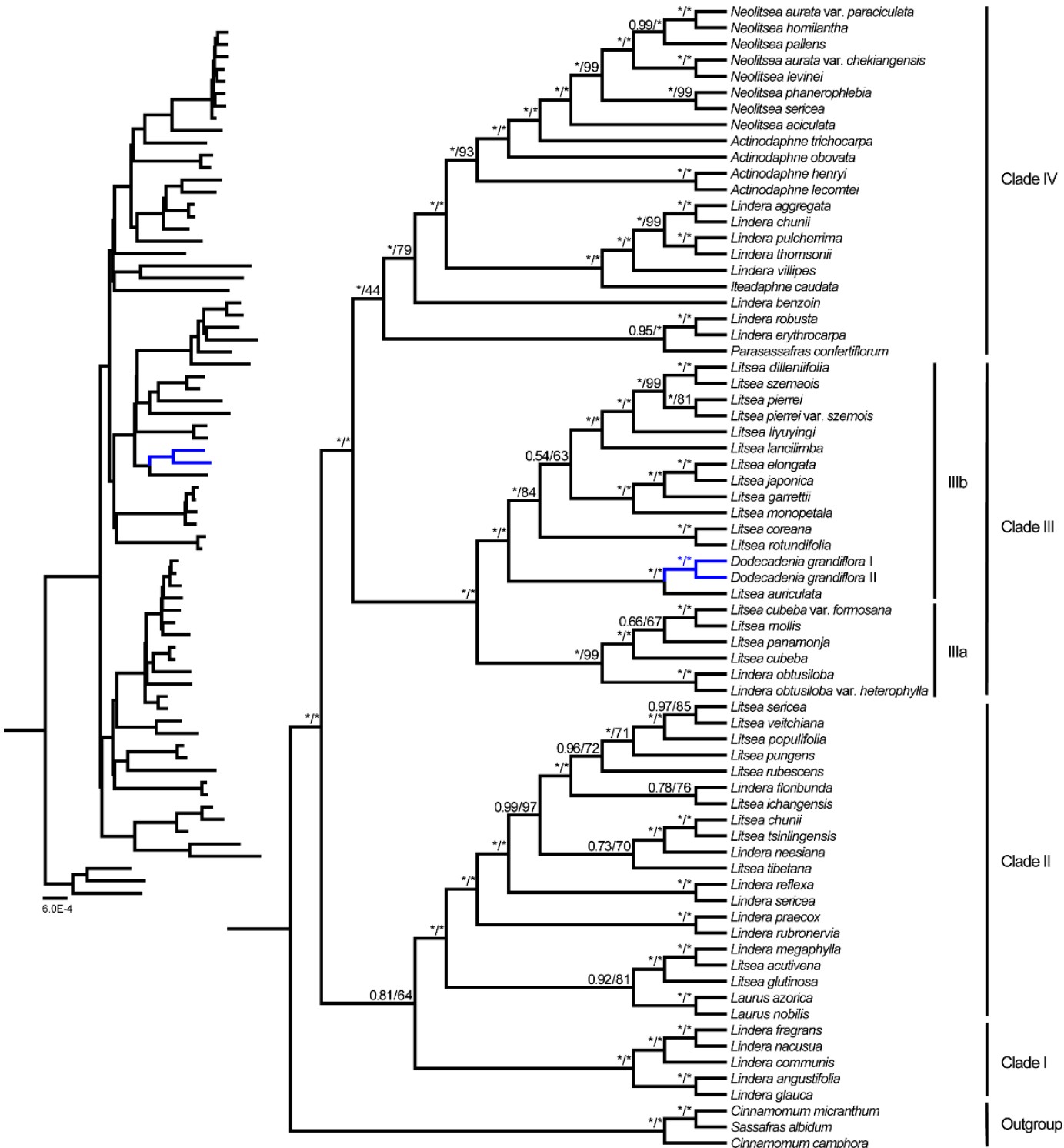

**Figure 6.** Complete plastome phylogenetic tree of *Litsea* complex inferred from Bayesian inference and maximum likelihood analysis. The support values (Bayesian inference/bootstrap value) are indicated at the branches. The star represents the value 1 or 100.

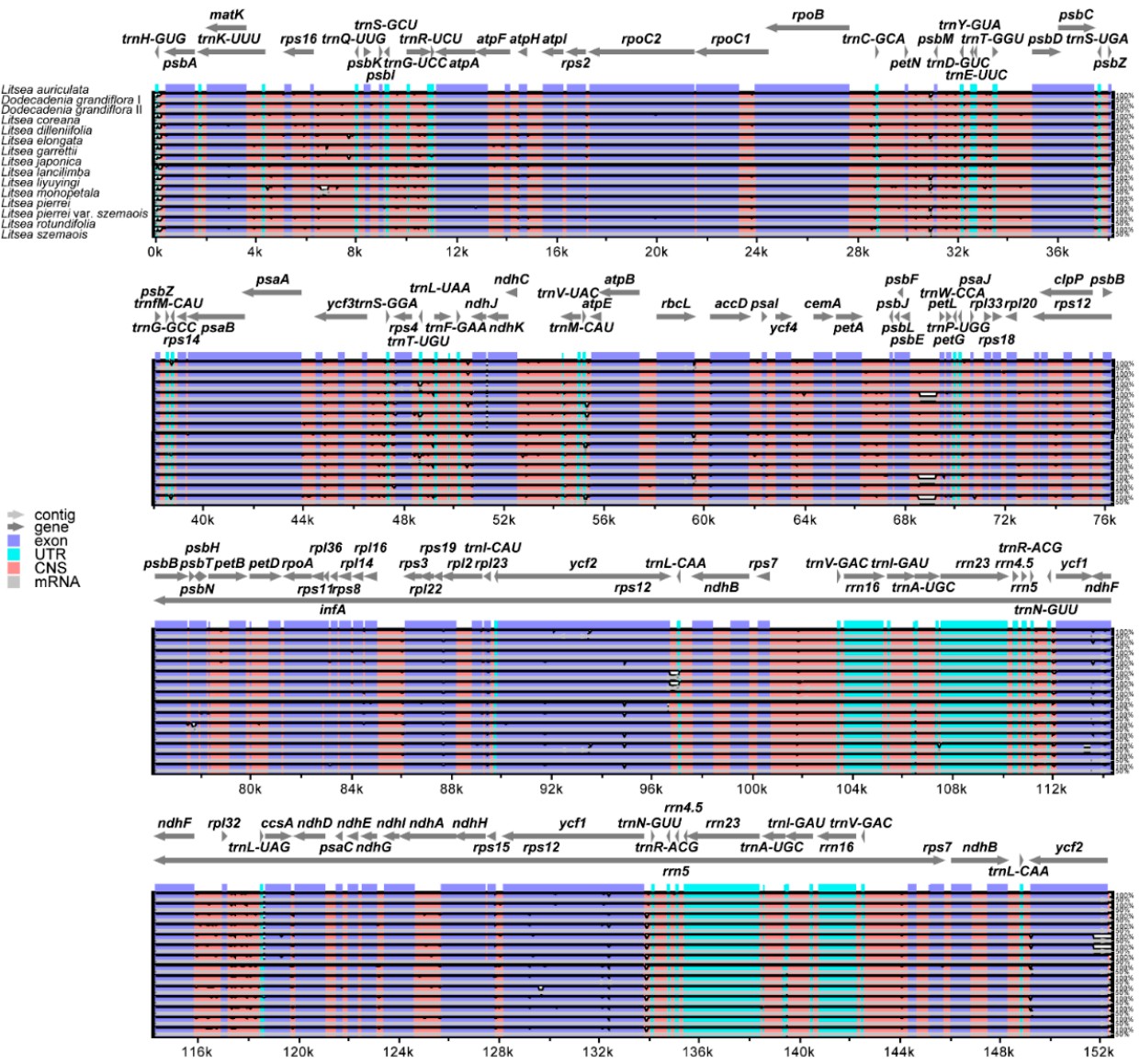

**Figure 7.** Visualization of *D. grandiflora* and 13 *Litsea* plastomes with *Litsea auriculata* as reference.

## 4. Discussion

Complete plastomes analyses demonstrated utility and simplicity in making inferences regarding systematics and phylogeny [6,15,17,18,36]. In the study, two complete plastomes of *D. grandiflora* were firstly reported, which broaden the knowledge about phylogeny of the *Litsea* complex. The genome sizes of the *D. grandiflora* examined here are similar to those of other Lauraceae [6,19,20,37]. The two newly assembled *D. grandiflora* plastomes size were 152,659 bp and 152,773 bp, and show some length variation in LSC, SSC and IR. Compared with *D. grandiflor* I, *D. grandiflor* II has three sequence inserts with lengths of 17, 18 and 28 bp, located in the intergenic region *trnT-psbD*, *rps16-trnQ*, and *rpoB-trnC* of LSC, and an insert of 96 bp located in the intergenic region *ndhF-rpl32* of SSC. Ten microinversions, 77 indels, and 238 substitutions were identified between the two *D. grandiflora* plastomes. A total of 128 (113 unique) genes were identified in *D. grandiflora*. For most plastomes of Lauraceae species examined so far, the number of genes was indicated as 127–134 (112–116 unique) [6,15,19,37–44]. A total of 84 PCGs were identified in *D. grandiflora*. The differences among the counts of genes in the Lauraceae species may be due to different annotation methods [20] or reference genome.

Like most other Lauraceae, the codons coding for leucine and for cysteine in *D. grandiflora* plastomes were the most and the least frequent [19,20]. About one-half of the codons (31/64) were used more frequently than expected with an RSCU value >1, and the preferred codons in *D. grandiflora* more frequently ended in A/U (29) than in G/C (2). These results are similar to those reports in other Lauraceae [19]. However, the count of preferred codons of *Ocotea* species ending in A/U than in G/C were 25 vs. six [20].

The high levels of sequence variations could be used as a potential molecular marker for population genetic diversity analysis [14,19,37,45,46]. The number of repeats has a positive correlation with the level of biological evolution, and a large number of repeats indicates the stronger stability of the plastome. In the study, 30/36 repeats (>30 bp) were found in two genomes of *D. grandiflora*, and mainly distributed in LSC and IRs, which was similar with other Lauraceae. Due to its analytical and highly polymorphic nature, SSR has been widely used to assess genetic diversity within species and their relatives [47]. Mononucleotide SSRs were the most abundant in the *D. grandiflora* plastomes, while A/T, AT/TA, AAT/TAA, and AAAT/ATTT repeats were most found in the mono-, di-, tri-, and tetranucleotide types, which showed A/T repeats were common in the plastomes [18]. This might be due to the AT-rich composition in the plastome. Consistent with most of other angiosperm plant, most of the SSRs were located in the LSC region compared to SSC and IR regions [14,18,47].

Generally, the different value range of nucleotide diversity depends on the number of species used or the region of the genome analyzed of land plants [48]. To develop accurate and cost-effective molecular markers for phylogenetic analysis, we analyzed Pi within different plastome regions of 15 (two *D. grandiflora* and 13 *Litsea* species) and 68 *Litsea* complex species. The average Pi values across the 15 and 68 Lauraceae plastomes vary from 0.0031 to 0.0038, which shows more species are associated with greater variability. By comparing the *Litsea* complex plastome, we confirmed that the IR regions are more conservative than the LSC and SSC. The highly variable loci *trnG* intron and *ycf3-trnS* were specific to the hypervariable region among *D. grandiflora* and 13 *Litsea* plastomes, which could be used in *D. grandiflora* phylogenetic analyses or as excellent candidate markers for population genetic and phylogenetic analyses. Eleven regions *psbK-psbI*, *psbI-trnS*, *trnS-trnG*, *trnC-petN*, *psbM-trnD*, *ycf2*, *ndhF*, *ndhF-rpl32*, *rpl32-trnL*, *rps15-ycf1*, and *ycf1* were identified as hypervariable loci (Pi $\geq$ 0.0095) at the species level among the *Litsea* complex, which could be employed for facilitating a better-resolved molecular phylogeny of *Litsea* complex species. The highly variable hotspots *psbM-trnD*, *ndhF*, *ndhF-rpl32*, *rpl32-trpL*, and *ycf1* also were identified in other *Litsea* complex studies [10,19,37,41]. Four different hypervariable loci (*psbA-trnH*, *ycf2*, *ndhH*, *trnL-ndhF*) were found in the *Ocotea* complex [20]. The highly variable regions detected by comparing complete plastomes may be used as markers for species identification and phylogenetic study.

Due to the significant polymorphism, constructing the phylogeny of plant species using the complete plastome might serve as excellent information for the identification of plant species. Plastome sequences have been widely used to reconstruct phylogenetic relationships among angiosperms. The phylogenetic relationships based on complete plastome were consistent with the Angiosperm Phylogeny Group IV system [49]. According to the phylogenetic tree, 71 *Litsea* complexes are clustered into four clades, which are supported with relatively high node support values. Earlier studies of core Lauraceae showed that the phylogenetic position of *D. grandiflora* has been controversial. Li et al. support *D. grandiflora* as the sister to *Laurus nobilis* or with a close relationship to several *Litsea* based on plastid gene or nuclear rDNA [11,12]. Our results support that *Lindera*, *Litsea*, and *Actinodaphne* were either poly- or paraphyletic [13]. *Dodecadenia grandiflora* is closely related to 12 species of *Litsea* and exhibited the closest relationship with *Litsea auriculata*. The phylogeny analysis based on the nrDNA sequences also showed a close relationship between *D. grandiflora* and the other 12 species of *Litsea* (unpublished data). Based on the above results, we suggested that the genus *Dodecadenia* could be reassigned to a new genus with 12 species of clade IIIb. This study provides significant insights into plastome

evolution and phylogenetic reconstruction of *Dodecadenia* species. Further studies using mitochondrial and nuclear datasets could help to understand the phylogenetic relationship of the *Litsea* complex.

## 5. Conclusions

This study firstly sequenced and assembled two complete plastomes of *D. grandiflora* from China. Two highly specific hypervariable loci, *trnG* intron and *ycf3-trnS*, were identified among *D. grandiflora* and related *Litsea* plastomes, which may be useful in species identification or as excellent candidate markers for population genetic and phylogenetic analyses. The phylogenetic analyses revealed *D. grandiflora* is closely related to 12 species of the *Litsea* genus. Our results will help improve our understanding of phylogenetics and provide substantial guidance for plastome engineering research of *D. grandiflora*.

**Supplementary Materials:** The supporting information can be downloaded at: https://www.mdpi.com/article/10.3390/f13081240/s1, Figure S1: Amino acids frequencies in the *Dodecadenia grandiflora* plastome protein-coding sequences; Figure S2: Mauve alignment of *Dodecadenia grandiflora* and 13 *Litsea* plastome revealing no interspecific rearrangements; Figure S3: Comparison of Pi values, (a) among *D. grandiflora* and 13 *Litsea* plastomes, and (b) among 68 plastomes of the *Litsea* complex; Table S1: List and accession number of 71 taxa used in this study; Table S2: Relative synonymous codon usage of *Dodecadenia grandiflora* plastoms; Table S3: RNA editing sites amino acid changes of *Dodecadenia grandiflora* chloroplast genomes; Table S4: Oligonucleotide repeats (>30 bp) identified within the plastoms of *Dodecadenia grandiflora*; Table S5: SSRs within the *Dodecadenia grandiflora* plastome.

**Author Contributions:** Conceptualization, Y.S. and C.L.; methodology, C.L. and J.C.; validation, C.L., L.H. and Y.S.; formal analysis, C.L., H.C. and Y.S.; resources, Y.S.; data curation, C.L., X.T. and Y.S.; writing—original draft preparation, C.L. and Y.S.; writing—review and editing, C.L., L.H., Y.S. and X.T.; funding acquisition, C.L., L.H. and Y.S. All authors have read and agreed to the published version of the manuscript.

**Funding:** This research and the APC was funded by the National Natural Science Foundation of China (grant no. 32060710, 32100010 to C.L. and L.H.) and Applied Basic Research Projects of Yunnan (grant no. 2019FB057 to Y.S.).

**Data Availability Statement:** The data presented in this study are available in the article and Supplementary Materials. The complete plastomes data of this study are openly available in GenBank of NCBI at https://www.ncbi.nlm.nih.gov; accession number: ON931229 and ON931230 (accessed on 3 July 2022).

**Conflicts of Interest:** The authors declare no conflict of interest.

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
