# Peer review of "Characteristics of the Complete Plastid Genome Sequences of the Monotypic Genus Dodecadenia (Family: Lauraceae) and Its Phylogenomic Implications"

_forests, doi:10.3390/f13081240_

Round 1

Reviewer 1 Report

The aim of this work is to perform both genomics and phylogenomics analysis on the plastidial DNA of Dodecadenia grandiflora. While I am not certain of the importance of studying this species in particular, the paper is a good piece of work, well documented and linear in its development, therefore I deem that is suited for publication after a minor revision specifically answering these few points:

- at line 114 the Authors say: “The sequencing adapters and low-quality reads were filtered”. Please provide both the coverage and the threshold used for filtering;

- starting at line 148 and throughout the whole manuscript: Pi is actually p;

- the legend of Figure 6 is lacking some information of interest, such as the meaning of the numbers at the nodes of the tree. Actually, the first should be the probability of that node estimated by Bayesian inference and the second the likelihood, I think. Better explain.

- last, please check the English thoroughly. As an example see lines 137-138: “These sequences were aligned using MAFFT v. 7 [31] and manually edited the matrices using BioEdit.” Many other syntax errors like this one can be found.

Author Response

Response to Reviewer

Dear Reviewer:

Thank you for your comments concerning our manuscript entitled “Characteristics of the Complete Plastid Genome Sequences of the Monotypic Genus Dodecadenia (Family: Lauraceae) and Its Phylogenomic Implications” (forests-1829766). Those comments are all valuable and very helpful for revising and improving our paper, as well as the important guiding significance to our researches. We have studied all comments carefully and have made corrections which we hope meeting the needs to get the approval for publication. Revised parts are marked up using the *Track Changes* function in the manuscript. The main corrections in the paper and the responds to the reviewer’s comments are as flowing:

Responds to the reviewer’s comments:

The aim of this work is to perform both genomics and phylogenomics analysis on the plastidial DNA of Dodecadenia grandiflora. While I am not certain of the importance of studying this species in particular, the paper is a good piece of work, well documented and linear in its development, therefore I deem that is suited for publication after a minor revision specifically answering these few points.

Response: Thank you for your good evaluation and some pertinent comments to our manuscripts.

1- at line 114 the Authors say: “The sequencing adapters and low-quality reads were filtered”. Please provide both the coverage and the threshold used for filtering;

Response: Thank you for your question. Illumina PE sequencing generated about 21,041,000 clean reads, with mean coverage 95 (X) of chloroplast genome. Quality control standard: When the N content in any sequencing read exceeds 10% of the base number of the read, remove the paired reads; When the low-quality (Q < =5) base number contained in any sequencing read exceeds 50% of the base number of the read, remove this paired reads.

2- starting at line 148 and throughout the whole manuscript: Pi is actually p;

Response: Thank you for your question. We checked the literature and found that nucleotide diversity statistic should be described as Pi.

3- the legend of Figure 6 is lacking some information of interest, such as the meaning of the numbers at the nodes of the tree. Actually, the first should be the probability of that node estimated by Bayesian inference and the second the likelihood, I think. Better explain.

Response: Thank you for your kind advice. We have added the legend information: The support values (bootstrap value/ Bayesian inference) are indicated at the branches.

4- last, please check the English thoroughly. As an example see lines 137-138: “These sequences were aligned using MAFFT v. 7 [31] and manually edited the matrices using BioEdit.” Many other syntax errors like this one can be found.

Response: Thank you for your kind comments. We have revised the relevant sentences.

Reviewer 2 Report

The manuscript reported on the plastid genome sequences of two accessions of Dodecadenia grandiflora, a species belonging to a monotypic genus of Lauraceae. Overall, the information provided is complete and the findings meet the objectives of the study. Language wise, an English check is still required for simple grammatical mistakes that can be found here and there. English is written in a mixture of British English and American English. In general, I will highlight on few important comments that the authors should give more attention at:

1.  The title uses the word "plastid"; hence, I expect instead of "chloroplast", the word "plastid" and "plastome" should be applied throughout the manuscript. The word "chloroplast" can still be found used extensively in the abstract and content. Please choose either "chloroplast" or "plastid".

2. There is no explanation on why two accessions are used for this study. Please provide an important reason why two accessions are required; is there anything you want to prove?

3. Line 70. What is ETS? the full term.

4. Line 87. The genus name should be spelled in full for the first word in a sentence. Check throughout.

5. Line 98. You cannot construct a phylogeny. You can only construct a phylogenetic tree.

6. Please provide the versions of the Bioinformatics tools used.

7. Hamming distance does not come with a "bp" unit.

8. How do you know that GTR+F+R2 is the optimum substitution model in IQTREE ML analysis? also, write in full for GTR+F+R2. and this is not a GAMMA model.

9. Line 183-198. The amino acids are not written in full at first occurrence in the text, except for few.

10. I have no idea the reason behind to conduct the sliding window analysis (DnaSP) showed in Figure 8. It makes nearly no sense to know the highly variable regions between the monotypic genus and other Litsea species. Although it was mentioned in the Discussion that the regions are potential markers for phylogenetic studies, but I do not see the importance to include the monotypic genus if you are just looking for a marker for Litsea. If there is no solid reason on conducting such analysis, please remove it from the study. It is more meaningful to provide the intraspecific distance between the two accessions, and also the difference in plastid genome of where the single nucleotide polymorphisms (SNPs) are located between the two accessions. Still, it comes back to the reason on why you need two accessions in this study.

11. Line 364. Suggesting a new nomenclature for Dodecadenia just by referring to the findings obtained from the phylogenetic tree showed in this study is not advisable. The authors did not provide any discussion on the placement of Dodecadenia based on the nuclear tree. If the nuclear tree has a different topology to that of the plastid tree, then hybridization is speculated to be present; thus, plastid tree is less accurate to identify the placement of Dodecadenia.

12. after reading to the end, it is unfortunate to say that Dodecadenia is not a (possible) sister to Litsea auriculata as Litsea is a big genus. having just less than half of the species included in the phylogenetic analysis cannot give you a definite idea who the is the sister species of Dodecadenia. They are just "closely related".

Author Response

Response to Reviewer

Dear Reviewer:

Thank you for your comments concerning our manuscript entitled “Characteristics of the Complete Plastid Genome Sequences of the Monotypic Genus Dodecadenia (Family: Lauraceae) and Its Phylogenomic Implications” (forests-1829766). Those comments are all valuable and very helpful for revising and improving our paper, as well as the important guiding significance to our researches. We have studied all comments carefully and have made corrections which we hope meeting the needs to get the approval for publication. Revised parts are marked up using the *Track Changes* function in the manuscript. The main corrections in the paper and the responds to the reviewer’s comments are as flowing:

Responds to the reviewer’s comments:

The manuscript reported on the plastid genome sequences of two accessions of Dodecadenia grandiflora, a species belonging to a monotypic genus of Lauraceae. Overall, the information provided is complete and the findings meet the objectives of the study. Language wise, an English check is still required for simple grammatical mistakes that can be found here and there. English is written in a mixture of British English and American English.

Response: Thank you for your good evaluation and some pertinent comments to our manuscripts.

  1. The title uses the word "plastid"; hence, I expect instead of "chloroplast", the word "plastid" and "plastome" should be applied throughout the manuscript. The word "chloroplast" can still be found used extensively in the abstract and content. Please choose either "chloroplast" or "plastid".

Response: Thank you for your kind comments. We have applied the word "plastid" and "plastome" instead of “chloroplast” and “chloroplast genome” throughout the manuscript.

  1. There is no explanation on why two accessions are used for this study. Please provide an important reason why two accessions are required; is there anything you want to prove?

Response: Thank you for your comments. In the study, we used two samples from Tibet and Yunnan, China, which can robustly support the phylogeny of Dodecadenia genus.

  1. Line 70. What is ETS? the full term.

Response: Thank you for your question. We have added the full term “the external transcribed spacer”.

  1. Line 87. The genus name should be spelled in full for the first word in a sentence. Check throughout.

Response: We have checked throughout the manuscript and spelled the genus name in full for the first word in a sentence.

  1. Line 98. You cannot construct a phylogeny. You can only construct a phylogenetic tree.

Response: We have revised the relevant sentences.

  1. Please provide the versions of the Bioinformatics tools used.

Response: We have added the versions of the Bioinformatics tools used.

  1. Hamming distance does not come with a "bp" unit.

Response: We have deleted “bp”.

  1. How do you know that GTR+F+R2 is the optimum substitution model in IQTREE ML analysis? also, write in full for GTR+F+R2. and this is not a GAMMA model.

Response: Thank you for your comments. The best substitution model (GTR+F+R2) was tested by AIC in IQ-TREE v.2.1.1. We have revised the relevant sentences.

  1. Line 183-198. The amino acids are not written in full at first occurrence in the text, except for few.

Response: We have added the full name of amino acids at first occurrence in the text.

  1. I have no idea the reason behind to conduct the sliding window analysis (DnaSP) showed in Figure 8. It makes nearly no sense to know the highly variable regions between the monotypic genus and other Litsea species. Although it was mentioned in the Discussion that the regions are potential markers for phylogenetic studies, but I do not see the importance to include the monotypic genus if you are just looking for a marker for Litsea. If there is no solid reason on conducting such analysis, please remove it from the study. It is more meaningful to provide the intraspecific distance between the two accessions, and also the difference in plastid genome of where the single nucleotide polymorphisms (SNPs) are located between the two accessions. Still, it comes back to the reason on why you need two accessions in this study.

Response: Thank you for your comments. The phylogeny showed a closely relationship between D. grandiflora and other 12 species of Litsea, so we wanted to know the nucleotide variability across the 15 plastomes, which might serve the identification of plant species. In view of the expert opinions, we showed the information of Figure 8 as supplement Figure 3. We also compared the differences of the SNPs in the two D. grandiflora plastomes (see Line 247-250, the last sentence of the second paragraph in part 3.6).

  1. Line 364. Suggesting a new nomenclature for Dodecadenia just by referring to the findings obtained from the phylogenetic tree showed in this study is not advisable. The authors did not provide any discussion on the placement of Dodecadenia based on the nuclear tree. If the nuclear tree has a different topology to that of the plastid tree, then hybridization is speculated to be present; thus, plastid tree is less accurate to identify the placement of Dodecadenia.

Response: Thank you for your kind comments. Based on the nrDNA sequences (unpublished data), we found D. grandiflora and other 12 species of Litsea kept a closely relationship, which can support the suggesting of a new nomenclature for Dodecadenia. And we have added the relevant discussion in our manuscript (Line 319-321).

  1. after reading to the end, it is unfortunate to say that Dodecadenia is not a (possible) sister to Litsea auriculata as Litsea is a big genus. having just less than half of the species included in the phylogenetic analysis cannot give you a definite idea who is the sister species of Dodecadenia. They are just "closely related".

Response: We have deleted the relevant sentences of sister relationship.
